# Functional outcomes after the treatment of hip fracture

Ai Takahashi[1,2]*, Hiroaki Naruse[2], Ippei Kitade[2], Seiichiro Shimada[2], Misao Tsubokawa[1,2], Yasuo Kokubo[1,2], Akihiko Matsumine[1,2]

1 Department of Orthopaedics and Rehabilitation Medicine, University of Fukui, Fukui Prefecture, Japan,
2 Division of Rehabilitation Medicine, University of Fukui Hospital, Fukui Prefecture, Japan

* aitun@u-fukui.ac.jp

**Data Availability Statement:** All relevant data are within the manuscript and its Supporting Information files (in Japanese).

**Funding:** The authors received no specific funding for this work.

## Abstract

Osteoporotic hip fracture is a major public health issue. Estimation of the outcome and maximization of functional recovery after fracture is very important in the treatment of older patients. The purposes of this study were to clarify the functional outcomes after the treatment of hip fracture and to identify the factors that influence functional recovery. In the present study, 228 patients admitted to an acute-care hospital from January 2016 to June 2018 were evaluated. The patients were categorized into a trochanteric fracture group (n = 128) and a neck fracture group (n = 100). We retrospectively reviewed their ambulation ability 6 months after fracture using the Functional Ambulation Category (FAC) score. The other survey items were the presurgical duration, length of hospital stay, time until beginning to walk using parallel bars, complications affecting treatment, and mortality rate. The 6-month follow-up rate was 54.4% (n = 124). The results showed that the patients with trochanteric fracture were significantly older than those with neck fracture (86 vs. 82 years, respectively; p = 0.03). In total, 85.0% of patients with trochanteric fracture and 92.2% of patients with neck fracture were independent ambulators before injury (FAC score of 4 or 5). The FAC score 6 months after fracture was positively correlated with the FAC score before fracture and at discharge (all p<0.001) and negatively correlated with patient age (p<0.001) and presurgical duration for patients with neck fracture (p = 0.04). There was no statistically significant correlation with the length of hospital stay or the time until beginning to walk using parallel bars. In conclusion, patients with trochanteric fractures were older than those with neck fractures. In both fracture types, walking recovery 6 months after hip fracture was related to the FAC score before injury and at discharge from an acute-care hospital but not to the time until beginning to walk using parallel bars.

## Introduction

Hip fracture is one of the most important health problems in patients of advanced age. Such fractures are classified as trochanteric fractures, neck fractures, and head fractures in the AO/OTA classification [1], and most osteoporosis-based hip fractures in patients of advanced age are trochanteric fractures (AO/OTA 31-A) and neck fractures (31-B). The incidence rate of

**Competing interests:** The authors have declared that no competing interests exist.

hip fracture increases with aging, and evaluation of the ratios of trochanteric and neck fractures has revealed that more neck fractures occur in patients aged <75 years and that more trochanteric fractures occur in patients aged >75 years [2, 3]. Early surgical treatment and remobilization are recommended in the international clinical guidelines [4]; however, conservative treatment based on traction is sometimes necessary for some patients when surgical treatment is not possible because of fragility, severe complications, or delayed discovery of the fracture. In addition, no specific protocol has been established for early rehabilitation of hip fractures. In particular, walking is sometimes started before the patient has sufficient basic physical and muscle strength because of strong concern about starting walking early after surgery. We hypothesized that functional recovery after hip fracture may not be related to the start of walking during the acute rehabilitation period.

Most patients with hip fracture are very old, and few reports have described treatment outcomes, including conservative treatment. In addition, the difference in treatment outcomes between trochanteric and neck fractures is unclear. Therefore, an understanding of the relatively short-term outcomes and the factors that influence functional recovery is clinically important. This study was performed to report the functional outcomes of trochanteric versus neck fractures including the patients received conservative treatment and associated factors 6 months after hip fracture.

## Methods

This was a retrospective cohort study. This research was approved by the Research Ethics Committee of University of Fukui (Permission number: 20190154). The data were corrected by medical records and analyzed anonymously. The patients included in the study were admitted to the University of Fukui Hospital, a 600-bed acute-care hospital located in the Hokuriku area of Japan, from January 2016 to June 2018. The study population comprised 228 patients (172 women and 56 men) categorized into the trochanteric fracture group (AO/OTA 31-A, n = 128) and the neck fracture group (31-B, n = 100). We evaluated the patients' ambulation ability before injury, at discharge, and 6 months after injury from the medical records using the Functional Ambulation Category (FAC) score [5]. The FAC is 6-point scale ranging from 0 (nonfunctional ambulator) to 5 (independent ambulator) that evaluates the ambulation status by determining how much human support the patient requires when walking. Other items evaluated in this study were the presurgical duration, length of hospital stay, time until beginning to walk using parallel bars, and complications affecting treatment, and mortality rate.

Differences between groups were examined using the Mann–Whitney U test for median age, median presurgical days, and median hospital days; the chi-squared test for sex and complications; and Spearman's correlation analysis for ambulation ability and correlating factors. A p value of <0.05 indicated a statistically significant difference between groups. All statistical analyses were performed using SPSS 10.0 (SPSS Inc., Chicago, IL, USA).

## Results

The patients' characteristics are shown in Table 1. All patients were divided into two groups; 128 had trochanteric fracture and 100 had neck fracture. The median age of all patients was 85 years (range, 32–99 years), and the patients with trochanteric fracture were significantly older than those with neck fracture (86 vs. 82 years, respectively; p = 0.03). Both types of fracture were more common in women (trochanteric fracture, 75.4%; neck fracture, 78.0%; p = 0.43). The main treatment for trochanteric fractures was osteosynthesis (83.6% of trochanteric fractures, n = 107), and the main treatment for neck fractures was bipolar hip arthroplasty (67.0% of neck fractures, n = 67). The numbers of patients treated conservatively were not

**Table 1. Patients' characteristics.**

|  | Total | Trochanteric fracture (AO/OTA 31-A) | Neck fracture (AO/OTA 31-B) | p value |
|---|---|---|---|---|
| Median age: years (range) | 85 (32–99) | 86 (32–99) | 82 (43–96) | 0.03 |
| Gender: n (%) |  |  |  |  |
| Men | 56 (24.6%) | 34 (26.6%) | 22 (22.0%) | 0.43 |
| Women | 172 (75.4%) | 94 (73.4%) | 78 (78.0%) | 0.43 |
| Treatment: n (%) |  |  |  |  |
| Osteosynthesis | 125 (54.8%) | 107 (83.6%) | 18 (18.0%) |  |
| Bipolar head arthroplasty | 69 (30.3%) | 2 (1.6%) | 67 (67.0%) |  |
| Conservative | 34 (14.9%) | 19 (14.8%) | 15 (15.0%) | 0.97 |
| Median presurgical days (range) | 7 (0–38) | 5 (0–31) | 8 (0–38) | <0.001 |
| Median hospital days (range) | 18 (2–114) | 16 (2–69) | 21 (8–114) | <0.001 |
| Complication: n (%) |  |  |  |  |
| Pneumonia | 21 (9.2%) | 11 (8.6%) | 10 (10.0%) | 0.72 |
| DVT/PE | 21 (9.2%) | 14 (10.9%) | 7 (7.0%) | 0.31 |
| Urinary infection | 14 (6.1%) | 9 (7.0%) | 5 (5.0%) | 0.55 |
| Diabetes | 18 (7.9%) | 11 (8.6%) | 7 (7.0%) | 0.66 |
| Necessity of presurgical drug management | 24 (10.5%) | 12 (9.4%) | 12 (12.0%) | 0.52 |
| Surgical site infection | 0 (0.0%) | 0 (0.0%) | 0 (0.0%) | 1 |
| Mortality: n (%) | 8 (6.5% of 124) | 6 (10.0% of 60) | 2 (3.1% of 64) | 0.12 |

Mann–Whitney U test for median age, median presurgical days, and median hospital days.

Chi-squared test for sex, complications, and mortality.

DVT: deep vein thrombosis.

PE: pulmonary embolism.

significantly different between the two fracture types (14.8% of patients with trochanteric fracture and 15.0% of those with neck fracture, p = 0.97). The median presurgical duration and median hospital period were longer in patients with neck fracture than in those with trochanteric fracture (5 vs. 8 days and 16 vs. 21 days, respectively; both p<0.01). The main presurgical problems were severe diabetes requiring control (7.9%) and anticoagulation drug management (10.5%). The total mortality rate was 6.5% (10.0% [n = 6] of patients with trochanteric fracture and 3.1% [n = 2] of those with neck fracture, p = 0.12). There was no significant difference in the presurgical complications and the total mortality rate.

Ambulation ability was assessed using the FAC score as shown in Fig 1. The 6-month follow-up rate was 54.4% (n = 124), and the main reason for drop-out was transfer in both groups. In total, 85.0% (n = 51) of patients with trochanteric fracture and 92.2% (n = 59) of those with neck fracture were independent walkers (FAC score of 4 or 5) before injury. Six months after fracture, 56.7% (n = 34) of patients with trochanteric fracture and 70.3% (n = 45) of those with neck fracture maintained their walking ability (p = 0.21). A total of 53.3% (n = 32) of patients with trochanteric fracture and 42.2% (n = 27) of those with neck fracture showed a decrease in their FAC score by ≥1 point (p = 0.21). The patients with trochanteric fracture were more likely to be nonfunctional ambulators or bed-ridden (FAC score of 0) than those with neck fracture (16.7% [n = 10] vs. 3.1% [n = 2], respectively; p = 0.011).

Fig 2 shows the factors correlated with the functional outcome. The FAC score at 6 months after fracture was positively correlated with the FAC score before fracture and at discharge (all p<0.001) and negatively correlated with patient age (p<0.001) and presurgical duration for

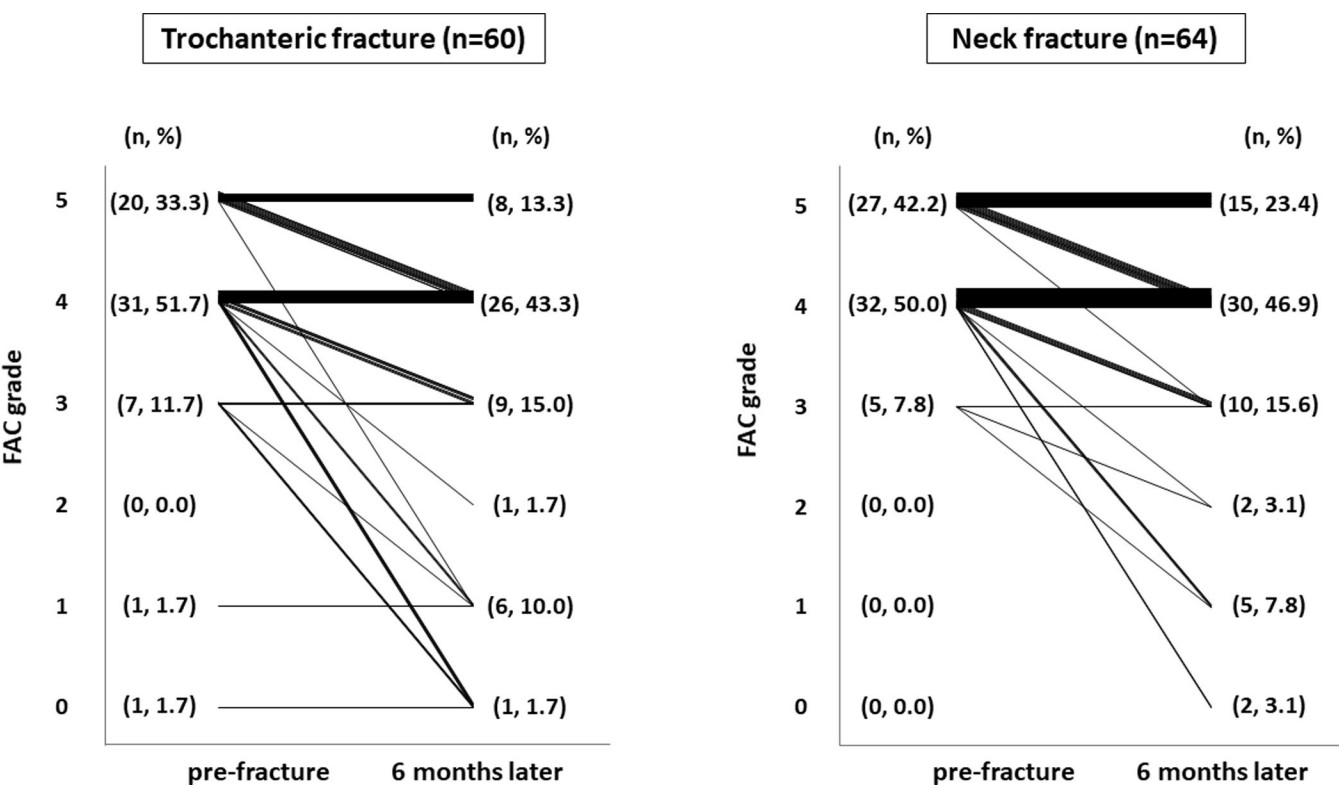

**Fig 1. FAC score before fracture and 6 months later.** The FAC score before fracture is shown on the left side of each graph, and the FAC score after 6 months is shown on the right side. The two scores are connected by a line, and the thickness of the line corresponds to the number of patients.

patients with neck fracture (p = 0.04). There was no statistically significant correlation with the presurgical duration for patients with trochanteric fracture (p = 0.65), length of hospital stay

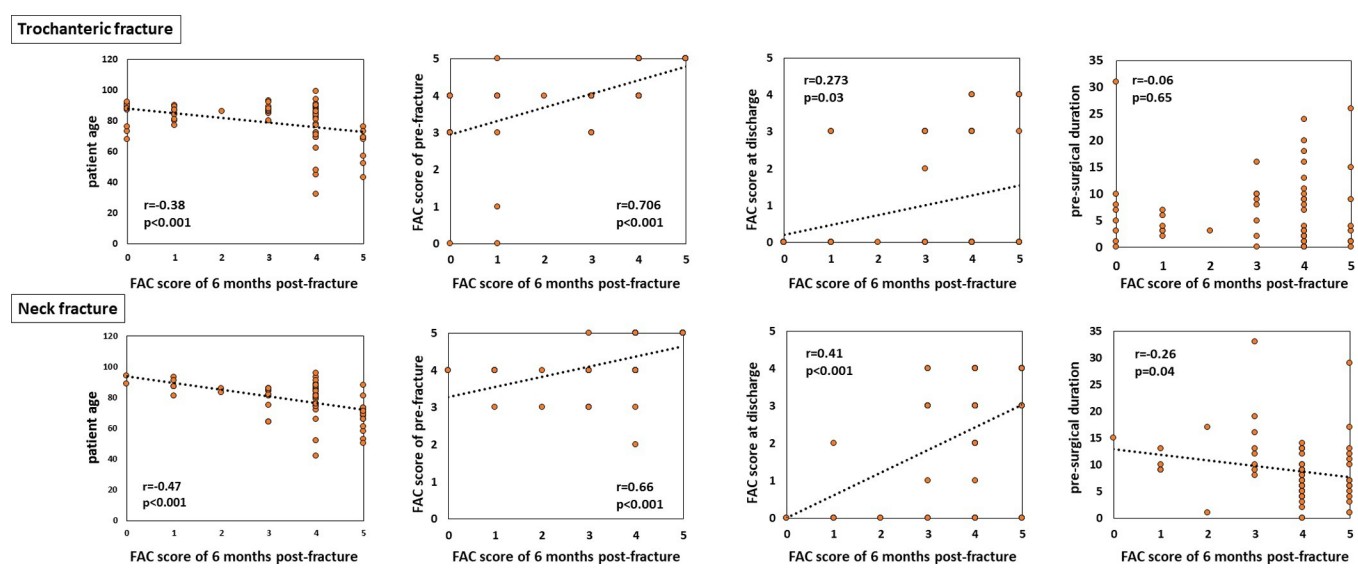

**Fig 2. Factors correlated with the functional outcome.**

(trochanteric fracture, p = 0.36; neck fracture, p = 0.15), or time until beginning to walk using parallel bars (trochanteric fracture, p = 0.30; neck fracture, p = 0.86).

## Discussion

Osteoporotic fracture is one of the most important medical/social problems leading to the need for long-term care and accounts for 12.5% of cases in which long-term care insurance is required [6]. In Japan, a nationwide survey by Orimo et al. [7] estimated that 37,600 men and 138,100 women sustained hip fractures in 2012 (total of 175,700 patients), and the annual number of patients is expected to increase in the future [8].

Osteoporotic hip fracture is divided into trochanteric fracture and neck fracture [1], and patients with trochanteric fracture are generally older than those with neck fracture [2, 3]. In the present study, patients with trochanteric fracture were significantly older than those with neck fracture; thus, our data support previous studies. Bone fragility of the trochanter region is considered to be a cause of trochanteric fractures in older people. Tanner et al. [9] reported that the types of hip fracture differ between men and women and that as women get older, they are more likely to sustain trochanteric fractures than are men. The authors considered that the intertrochanteric region absorbs the force passed along to the neck of the femur and that women are more likely to develop trochanteric fractures because they are more prone to osteoporosis than men [9].

International guidelines recommend early surgical treatment and rehabilitation; however, conservative treatment is chosen for some patients because of pre-existing disease such as heart failure, respiratory disorders, diabetes, renal failure, and other conditions. In this study, a relatively high percentage of patients were selected for conservative treatment because many of the patients had been referred from other hospitals, and some of them were judged as having high anesthetic risk. The patients who received conservative treatment were transferred to another hospital and underwent protective care and rehabilitation at that institution. In the present study, patients aged >85 years accounted for about 50% of the total patients, and 45% of them were >90 years old; this is considered to be the reason for the relatively high 6-month mortality rate (6.5%) and proportion of nonfunctional ambulators or bed-ridden patients (9.7%). However, some selection bias may have occurred because we excluded patients who did not present to our hospital. The follow-up rate were relatively low because the many patients returned to the home town far from our hospital, and were supported only by local facility care services. In previous studies that evaluated treatment outcomes including those for patients who received conservative treatment, the annual mortality rate ranged from 10% to 40% [10–12]. Factors reportedly associated with higher mortality included aging, male sex, cognitive dysfunction, cardiovascular disease, respiratory disease, diabetes mellitus, and malignant tumors [10–15]. The Charlson comorbidity index [12] and the American Society of Anesthesiologists Physical Status Classification System [13] are were also both reportedly associated with mortality.

The functional prognosis of hip fractures differs between surgical and conservative treatment, and few reports have described treatment outcomes, including conservative treatment. In the present study, 56.7% of patients with trochanteric fracture and 70.3% of those with neck fracture maintained their walking ability at 6 months after fracture, and patients with trochanteric fracture were more likely to be nonfunctional ambulators or bed-ridden than those with neck fracture. Patient age may have been a confounding factor. Factors associated with the functional prognosis were patient age, the FAC score before fracture and at discharge, and presurgical days in patients with neck fracture. We found no correlation between presurgical days and hospital days. Previous reports have shown a strong association between functional

recovery and age, preoperative physical function, and cognitive function [16]. The cutoff value for age is not clear, but older age is associated with poorer recovery of walking ability. The motor Functional Independence Measure score [17] and the New Mobility Score [18] are examples of methods used to evaluate physical function. The FAC is a simple evaluation method, and the preoperative FAC score is related to the 6-month postoperative score. This scoring method is considered suitable for evaluating the walking ability of patients with proximal femoral fractures. Although we did not statistically analyze cognitive function in this study, cognitive function is evaluated in almost all patients, and occupational therapy is performed to maintain cognitive function and improve activities of daily living. About the timing of surgery, the National Institute for Health and Care Excellence recommends early surgery within 48 hours [4]; this strategy is associated with advantages such as reduced complications and improved functional recovery. Although early surgery is reported to be positively associated with the life prognosis [14], it is generally possible that a patient with no or few complications has undergone early surgery, and the effect of bias may be considered. In a Japanese study, 2010–2014 data showed that only 22.5% of patients underwent surgery within 2 days of hospitalization, and the risk of pneumonia and pressure ulcers was significantly reduced in the early surgery group [19].

Preoperative rehabilitation and early mobility are recommended, and there are numerous reports of valid rehabilitation protocols [20]. However, the 2011 Cochrane Review concludes that there is insufficient evidence from randomized trials to establish the best strategies for enhancing mobility after hip fracture surgery [21]. In the present study, the duration of time until beginning to walk using parallel bars was not related to the walking ability after surgical treatment, and we found that early compelled walking did not improve functional ability. Walking is unstable, slow, and poorly coordinated in most people of advanced age; this is caused by not only musculoskeletal weakness but also cardiovascular dysfunction and neurological problems or cognitive dysfunction [22]. Rehabilitation programs to regain ambulatory ability after hip fracture should include basic range-of-motion exercises, muscular strengthening, aerobic exercise, and occupational therapy. Notably, however, the results of recent randomized controlled trials have indicated the beneficial effects of multidisciplinary rehabilitation and post-discharge exercise programs [23–26]. Based on these reports, we consider that not only acute treatment but also home exercise after discharge and a multidisciplinary approach are important for functional recovery and improvements in activities of daily living after hip fracture. In recent years in Japan, community activities have been vigorously conducted for the purpose of long-term care prevention. Therefore, proactive introduction of such services to patients with hip fracture can be expected to maintain and improve motor function.

## Conclusion

In conclusion, we found patients with trochanteric fractures were older than those with neck fractures, which supports the findings of previous studies. At least in our sample, walking recovery 6 months after hip fracture was related to the FAC score before injury and at discharge from an acute-care hospital but not to the time until beginning to walk using parallel bars in both fracture types. Walking ability at the time of discharge from an acute-care hospital can be a predictor of the outcome, but inappropriate early initiation of walking is not recommended.

## Supporting information

**S1 File. Anonymous patient data.**
(PDF)

## Acknowledgments

We thank Angela Morben, DVM, ELS from Edanz Group for editing a draft of this manuscript.

## Author Contributions

**Conceptualization:** Yasuo Kokubo.

**Data curation:** Ai Takahashi.

**Formal analysis:** Ai Takahashi, Hiroaki Naruse.

**Investigation:** Hiroaki Naruse, Ippei Kitade, Seiichiro Shimada.

**Supervision:** Misao Tsubokawa, Yasuo Kokubo, Akihiko Matsumine.

**Writing – original draft:** Ai Takahashi.

**Writing – review & editing:** Akihiko Matsumine.

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
