## [Decision Letter · Decision Letter 0]

24 Mar 2020

PONE-D-20-02630

Functional outcomes after the treatment of hip fracture

PLOS ONE

Dear Dr. Takahashi,

Thank you for submitting your manuscript to PLOS ONE. After careful consideration, we feel that it has merit but does not fully meet PLOS ONE’s publication criteria as it currently stands. Therefore, we invite you to submit a revised version of the manuscript that addresses the points raised during the review process.

The authors are required to identify their primary objective of the study and to other secondary objectives, then they can link their methodology and results relative to the their objectives.

We would appreciate receiving your revised manuscript by May 08 2020 11:59PM. To enhance the reproducibility of your results, we recommend that if applicable you deposit your laboratory protocols in protocols.io, where a protocol can be assigned its own identifier (DOI) such that it can be cited independently in the future. For instructions see: http://journals.plos.org/plosone/s/submission-guidelines#loc-laboratory-protocols

We look forward to receiving your revised manuscript.

Kind regards,

Osama Farouk

Academic Editor

PLOS ONE

Additional Editor Comments (if provided):

The article needs extensive revision of the points raised by the reviewers. Please, respond to the reviewers' comments one by one.

Journal Requirements:

2. Please ensure you have thoroughly discussed any potential limitations of this study within the Discussion section, for example the potential impact of confounding factors.

3. Please include your tables as part of your main manuscript and remove the individual files. Please note that supplementary tables (should remain/ be uploaded) as separate "supporting information" files

Reviewers' comments:

Reviewer's Responses to Questions

**Comments to the Author**

1. Is the manuscript technically sound, and do the data support the conclusions?

Reviewer #1: Yes

Reviewer #2: No

2. Has the statistical analysis been performed appropriately and rigorously? 

Reviewer #1: Yes

Reviewer #2: No

3. Have the authors made all data underlying the findings in their manuscript fully available?

Reviewer #1: Yes

Reviewer #2: No

4. Is the manuscript presented in an intelligible fashion and written in standard English?

Reviewer #1: Yes

Reviewer #2: No

5. Review Comments to the Author

Reviewer #1: Introduction:

Sentences too long.

Methods:

What were the medical conditions prior to trauma? Was there a significant difference between trochanteric and neck fractures regarding pre-existing disease/comorbidities?

Please clarify why the pre-surgical duration was that long in both groups. In Germany, we are obligated to perform surgery within 36 hours in hip fractures.

Did your patients stay in an acute trauma facility for over 18 days?

Please explain the high rates of conservative treatment. Even though only 14 patients had a FAC of three or less, 34 patients were treated conservatively. I can hardly remember a geriatric/older patient in my clinic who was not surgically treated for a hip fracture.

How were the “nonfunfuctional or bed-ridden patients” treated?

Did your patients receive rehabilitation programs after being discharged? Did these programs have an effect?

Did the pre-operative ASA score have a predictive value of the post-surgical result?

Discussion:

Line 169pp: “We found that not only acute care and rehabilitation but also home exercise after

170 discharge and adequate social support are also important for functional recovery and improvements in activities of daily living after hip fracture.“

I cannot find any correlation in your text.

What is the reason for your high rates of mortality and non-walkers? Please discuss.

Reviewer #2: General comments:

The study idea about Functional outcomes after the treatment of hip fracture

is a clinically important area of research in hip fracture patients.

But unfortunately, the manuscript is not written in a way that made integration between the manuscript sections. There is discrepancy between the aim of the study and the methodology. Some linguistic revision is required.

However, I have provided some remarks below.

Abstract:

The abstract all over its section should be corrected accordingly after rewriting of the manuscript.

Introduction

- Replace cervical fracture with neck fracture to standardize.

- Classification of fractures indicate that it has an important significance in aim, methods and results.

- Between lines 57 to 62: is not related to the aim of the study.

- The introduction should concentrate on the factors affecting the functional outcomes and mortality after treatment of hip fracture, which is the point of the study.

- The aim of the study is different from the way of the methodology and results, The aim could be “ to study the difference in functional outcomes after treatment of neck and trochanteric types of hip fracture”

Methods:

- There is no mentioning to the study design.

- There was a clear statement in the methodology section in the abstract about inclusion of the sample as two categories” trochanteric and neck fractures”, but here it’s not clear. As it’s clear, this is the start point of data collection.

- The mean age is better to be mentioned in the results section.

- Mortality was not mentioned as an important finding in the follow up. Was mentioned in the abstract.

- Which data were exactly retrieved from the patients’ records and which were from the actual follow up? Or all data were retrieved from the records? Lines 76 and 77

- Were the data tested for normal distribution or not?

- In data analysis, what about tests of significance for non parametric data and presentation of data as median in spite of mean?

- Why multiple analysis was not done ?

Results:

- In general, No titles were wrote for tables and figures

- Put symbols inside tables and refer to the test used as a footnote under the tables.

- The way of presentation in results started with the categorization of the total sample to two fracture groups till the end of results.

- Table (1):

• Present age and duration as mean + SD or as median according to the normal parametric distribution of data.

• Present the age as range, there is age of 32years old, please analyze age in details as it’s an important factor

• What is “PE”? write in details.

• Please, mention all items that were mentioned in this table, write them in the methods section.

• Where is the mortality in both groups? Please include it. In addition to the mean duration of postoperative mortality duration.

• mention the drop out on both groups

- Figure (2):

• In page 12, lines 112 to 117: mention the drop out at the end of follow up.

•

Discussion:

- Page 13, Lines 121 to 136: Introduction not related to the findings of the study, just one short paragraph about the importance of the study.

- Present the findings of this study and then compare with the others’ results, explain??

- Here, other studies’ results were presented firstly.

- 1Page 14, lines 47 – 159: not related to the studied aspects.

- Page 14, sentence started in line 169 “we found that…” This is not studied in this study??

Conclusion:

- The mortality is not a primary outcome to be mentioned first here.

- Conclusion not based on the categorization of the study sample according to the type of fracture, and the difference in functional outcomes and mortality

- Include a recommendation in this section.

6. PLOS authors have the option to publish the peer review history of their article (what does this mean?). If published, this will include your full peer review and any attached files.

Reviewer #1: No

Reviewer #2: No

---

## [Author Response · Author response to Decision Letter 0]

8 May 2020

Point-by-Point Responses to Comments of Reviewer #1

We would like to thank Reviewer #1 for evaluating our manuscript. Our responses to the reviewer’s comments are provided below. We apologize for the statistical errors in the first manuscript. The data did not follow a normal distribution; therefore, we have corrected the median patient age, preoperative duration, and hospitalization duration and revised the statistical methods. 

1. What were the medical conditions prior to trauma? Was there a significant difference between trochanteric and neck fractures regarding pre-existing disease/comorbidities?

In lines 92 to 97 of the revised manuscript, we have explained that the patients had multiple complications but that there was no significant difference between the two fracture groups.

2. Please clarify why the pre-surgical duration was that long in both groups. In Germany, we are obligated to perform surgery within 36 hours in hip fractures.

As described in the text, there were many serious cases in our hospital in which the patients had been transferred/referred from other hospitals. In addition, some patients required preoperative anticoagulation therapy, and evaluation and treatment required a period of several days or more before surgery.

3. Did your patients stay in an acute trauma facility for over 18 days?

We apologize for the statistical error. As shown in Table 1, the median length of stay for all patients was 18 days. Almost all patients were transferred to another hospital for the purpose of continuing medical care and rehabilitation after treatment. Acute rehabilitation was performed in our hospital.

4. Please explain the high rates of conservative treatment. Even though only 14 patients had a FAC of three or less, 34 patients were treated conservatively. I can hardly remember a geriatric/older patient in my clinic who was not surgically treated for a hip fracture.

As mentioned in our response to Comment 2, many high-risk patients were transferred from other hospitals; this resulted in a high number of patients who needed conservative treatment at our hospital.

5. How were the “nonfunfuctional or bed-ridden patients” treated?

Did your patients receive rehabilitation programs after being discharged? Did these programs have an effect?

As mentioned in our response to Comment 3, almost all patients were transferred to another hospital for the purpose of continuing medical care and rehabilitation after treatment. Acute rehabilitation was performed in our hospital. Unfortunately, data could not be collected for some patients who were transferred to another hospital, which resulted in a low follow-up rate of 54.4%. The local newspapers and other sources provided fairly accurate information about the patients who died after discharge.

6. Did the pre-operative ASA score have a predictive value of the post-surgical result?

Some patients were judged to have a high surgical risk without consulting with the anesthesia department regarding the patient’s medical condition, and the ASA score was not present in the medical record. Such patients received conservative treatment. Therefore, we did not compare ASA scores with functional outcomes in this study.

7. Line 169pp: “We found that not only acute care and rehabilitation but also home exercise after discharge and adequate social support are also important for functional recovery and improvements in activities of daily living after hip fracture.“

I cannot find any correlation in your text.

We apologize for our unclear description. In lines 193 to 195, we have revised the text as follows: “Based on these previous reports, we consider that not only acute care and rehabilitation but also home exercise after discharge and adequate social support are important for functional recovery and improvements in activities of daily living after hip fracture.” As you pointed out, this content is not directly related to our results; however, we believe that continuation of exercise after discharge and enhancement of social security are important for hip fracture management. 

8. What is the reason for your high rates of mortality and non-walkers? Please discuss.

As mentioned above, many high-risk patients were treated in our hospital; as a result, 14.9% of patients received conservative treatment. Most patients who received conservative treatment were unable to walk, including those who did not die. However, there was no significant difference in mortality between the two groups. We believe that the reason for the higher number of nonfunctional ambulators among patients with trochanteric fractures is that age may have been a confounding factor. We have mentioned this in lines 164 to 166. 

 

Point-by-Point Responses to Comments of Reviewer #2

We thank Reviewer #2 for evaluating our manuscript. We have revised the content according to the advice provided. Our responses to the reviewer’s comments are provided below. 

Introduction:

1. Replace cervical fracture with neck fracture to standardize.

We have revised the text accordingly.

2. Classification of fractures indicate that it has an important significance in aim, methods and results.

As you mentioned, classification of the two fracture types is important in this study. We have revised the Introduction accordingly.

3. Between lines 57 to 62: is not related to the aim of the study.

We have removed this section of text.

4. The introduction should concentrate on the factors affecting the functional outcomes and mortality after treatment of hip fracture, which is the point of the study.

In this study, we focused on clarifying the functional outcome, including patients who received conservative treatment. Therefore, we have mentioned in the Introduction that there are few similar studies (line 56-58). Factors affecting the functional prognosis are discussed in the Discussion section (line 166-169).

5. The aim of the study is different from the way of the methodology and results, The aim could be “ to study the difference in functional outcomes after treatment of neck and trochanteric types of hip fracture”

We have revised the text accordingly. The end of the Introduction now reads, “Most patients with hip fracture are very old, and few reports have described treatment outcomes, including conservative treatment. In addition, the difference in treatment outcomes between trochanteric and neck fractures is unclear. Therefore, an understanding of the relatively short-term outcomes and the factors that influence functional recovery is clinically important. This study was performed to report the functional outcomes of trochanteric versus neck fractures and associated factors 6 months after hip fracture.”

Methods:

6. There is no mentioning to the study design.

We apologize for not mentioning the study design. At the beginning of the Methods section, we have added the following text: “This study was conducted by retrospective medical record evaluation.”

7. There was a clear statement in the methodology section in the abstract about inclusion of the sample as two categories” trochanteric and neck fractures”, but here it’s not clear. As it’s clear, this is the start point of data collection.

In accordance with your comment, we have added the following text to the revised manuscript: “The study population comprised 228 patients (172 women and 56 men) categorized into the trochanteric fracture group (AO/OTA 31-A, n=128) and the neck fracture group (31-B, n=100).”

8. The mean age is better to be mentioned in the results section.

We have accordingly mentioned patient age in the Results section.

9. Mortality was not mentioned as an important finding in the follow up. Was mentioned in the abstract.

We have revised the abstract in accordance with your comment.

10. Which data were exactly retrieved from the patients’ records and which were from the actual follow up? Or all data were retrieved from the records? 

At the beginning of the Methods section, we have mentioned that all data were collected from the medical records.

11. Were the data tested for normal distribution or not?

We apologize for our statistical error in the first manuscript. The data did not follow a normal distribution. Therefore, we have corrected the median patient age, preoperative duration, and hospitalization duration and changed the statistical method.

12. In data analysis, what about tests of significance for non parametric data and presentation of data as median in spite of mean?

We changed the statistical method in accordance with your advice. Differences between groups were examined using the Mann–Whitney U test for median age, median presurgical days, and median hospital days.

13. Why multiple analysis was not done ?

Although not described in the text, no statistically significant difference was found in the factors associated with the functional outcome by multivariate analysis.

Results:

14. In general, No titles were wrote for tables and figures

We have added titles to the tables and figures.

15. Put symbols inside tables and refer to the test used as a footnote under the tables.

We have revised the tables according to your instructions.

16. The way of presentation in results started with the categorization of the total sample to two fracture groups till the end of results.

We have accordingly mentioned the categorization of the patients at the beginning of the Results section.

Table (1):

17. Present age and duration as mean + SD or as median according to the normal parametric distribution of data.

The data are now presented as median values.

18. Present the age as range, there is age of 32 years old, please analyze age in details as it’s an important factor

We have accordingly presented the patients’ age as a range.

19. What is “PE”? write in details.

“PE” stands for pulmonary embolism. We apologize for omitting this definition; it has been added to Table 1.

20. Please, mention all items that were mentioned in this table, write them in the methods section.

We have mentioned all items from the table in the Methods section (line 68-74).

21. Where is the mortality in both groups? Please include it. In addition to the mean duration of postoperative mortality duration.

We have accordingly added the mortality rates for each group to Table 1. Because of the method of data acquisition, we were unable to obtain the accurate average mortality period.

22. mention the drop out on both groups

We have stated that the reason for drop-out was transfer in both groups.

Figure (2):

23. In page 12, lines 112 to 117: mention the drop out at the end of follow up.

At the beginning of the Methods section, we have stated that all data in this study were obtained from the patients’ medical records (line 64). Therefore, we have omitted all data that were not obtained from the medical records.

Discussion:

24. Page 13, Lines 121 to 136: Introduction not related to the findings of the study, just one short paragraph about the importance of the study.

We think epidemiological description in Japan is important; the unrelated text has been deleted.

25. Present the findings of this study and then compare with the others’ results, explain??

We have revised the Discussion section to compare our results with past reports.

26. Here, other studies’ results were presented firstly.

We prefer to mention the results of other studies first, if possible. We have described our results and others in the same paragraph.

27. Page 14, lines 47 – 159: not related to the studied aspects.

In the final paragraph, we discuss rehabilitation and integrated management. We have listed the types of rehabilitation that are reportedly useful for hip fractures, and we consider that both acute rehabilitation and long-term support are necessary.

28. Page 14, sentence started in line 169 “we found that…” This is not studied in this study??

We apologize for our unclear description. In lines 193 to 195, we have revised the text as follows: “Based on these previous reports, we consider that not only acute care and rehabilitation but also home exercise after discharge and adequate social support are important for functional recovery and improvements in activities of daily living after hip fracture.” As you pointed out, this content is not directly related to our results; however, we believe that continuation of exercise after discharge and enhancement of social security are important for hip fracture management.

Conclusion:

29. The mortality is not a primary outcome to be mentioned first here.

As you pointed out, mortality is not the main outcome; therefore, we did not list it here.

30. Conclusion not based on the categorization of the study sample according to the type of fracture, and the difference in functional outcomes and mortality

We have revised the Conclusion section as follows: “In conclusion, we found patients with trochanteric fractures were older than those with neck fractures, which supports the findings of previous studies. At least in our sample, walking recovery 6 months after hip fracture was related to the FAC score before injury and at discharge from an acute-care hospital but not to the time until beginning to walk using parallel bars in both fracture types.”

31. Include a recommendation in this section.

We have added the following sentence to the end of the Conclusion section: “Walking ability at the time of discharge from an acute-care hospital can be a predictor of the outcome, but inappropriate early initiation of walking is not recommended.”

---

## [Decision Letter · Decision Letter 1]

26 May 2020

PONE-D-20-02630R1

Functional outcomes after the treatment of hip fracture

PLOS ONE

Dear Dr. Takahashi,

Thank you for submitting your manuscript to PLOS ONE. After careful consideration, we feel that it has merit but does not fully meet PLOS ONE’s publication criteria as it currently stands. Therefore, we invite you to submit a revised version of the manuscript that addresses the points raised during the review process.

We look forward to receiving your revised manuscript.

Kind regards,

Osama Farouk

Academic Editor

PLOS ONE

Additional Editor Comments (if provided):

The authors are required to respond to all reviewer's comments. The discussion needs extensive revision.

Reviewers' comments:

Reviewer's Responses to Questions

**Comments to the Author**

1. If the authors have adequately addressed your comments raised in a previous round of review and you feel that this manuscript is now acceptable for publication, you may indicate that here to bypass the “Comments to the Author” section, enter your conflict of interest statement in the “Confidential to Editor” section, and submit your "Accept" recommendation.

Reviewer #1: All comments have been addressed

Reviewer #2: (No Response)

2. Is the manuscript technically sound, and do the data support the conclusions?

Reviewer #1: Yes

Reviewer #2: (No Response)

3. Has the statistical analysis been performed appropriately and rigorously? 

Reviewer #1: Yes

Reviewer #2: (No Response)

4. Have the authors made all data underlying the findings in their manuscript fully available?

Reviewer #1: Yes

Reviewer #2: (No Response)

5. Is the manuscript presented in an intelligible fashion and written in standard English?

Reviewer #1: Yes

Reviewer #2: (No Response)

6. Review Comments to the Author

Reviewer #1: Dear authors,

thank you for revising your paper.

To my mind your paper is now technical Sound and all data to Support your conclusions is provided. Statistical analyses have improved.

Reviewer #2: Overall, the changes that were made in response to the review are accepted except:

1- Point 6 in methodology, “ retrospective medical record evaluation” this is not a study design, the study design of this research is “ retrospective cohort study”.

2- Discussion:

Presentaion of the discussion should be done with the study presentation first then to compare with other studies, not the reverse.

There is no discussion for morbidity or mortality factors

The age and sex factors are discussed well.

No discussion with the hospital stay,……

No discussion with factors related to the walking ability.

In pages 17& 18, lines 240 to 262: are not related to presented data. In this study there are no evaluation of the rehabilitation done for patients in both groups. Could be written in two to three lines only.

Write recommendation related to results.

7. PLOS authors have the option to publish the peer review history of their article (what does this mean?). If published, this will include your full peer review and any attached files.

Reviewer #1: No

Reviewer #2: No

---

## [Author Response · Author response to Decision Letter 1]

30 Jun 2020

Point-by-Point Responses to Comments of Reviewer #1

We would like to thank Reviewer #1 for evaluating our manuscript. We added discussions about mortality, length of presurgical duration, and factors associating with walking ability, resulting in changes from Discission lines 150 to 161, 178 to 185, and lines 167 to 178. Also, the description of rehabilitation was changed by citing some recent randomized controlled trials (lines 186 to 189 and 196 to 201).

 

Point-by-Point Responses to Comments of Reviewer #2

We thank Reviewer #2 for evaluating our manuscript. We have revised the content according to the advice provided. Our responses to the reviewer’s comments are provided below. 

1. Point 6 in methodology, “ retrospective medical record evaluation” this is not a study design, the study design of this research is “ retrospective cohort study”.

We have revised the text accordingly (line 63 to 65).

Discussion:

2. Presentaion of the discussion should be done with the study presentation first then to compare with other studies, not the reverse.

We have revised the text accordingly.

3. There is no discussion for morbidity or mortality factors

We added the discussion about mortality rate in lines 150 to 161.

4. No discussion with the hospital stay,……

We added the discussion about presurgical duration in lines 178 to 185. In this study, there was no correlation wit hospital stay and functional recovery, so we are not discussing length of hospital stay.

5. No discussion with factors related to the walking ability.

We added the discussion about factors related to the walking ability in lines 167 to 178.

6.In pages 17< 18, lines 240 to 262: are not related to presented data. In this study there are no evaluation of the rehabilitation done for patients in both groups. Could be written in two to three lines only. Write recommendation related to results.

We omitted the item on rehabilitation content and revised it by citing some recent randomized controlled trials. (lines 186 to 189 and 196 to 201).

---

## [Decision Letter · Decision Letter 2]

13 Jul 2020

Functional outcomes after the treatment of hip fracture

PONE-D-20-02630R2

Dear Dr. Takahashi,

We’re pleased to inform you that your manuscript has been judged scientifically suitable for publication and will be formally accepted for publication once it meets all outstanding technical requirements.

Kind regards,

Osama Farouk

Academic Editor

PLOS ONE

Additional Editor Comments (optional):

All reviewers' comments were addressed.

Reviewers' comments:

Reviewer's Responses to Questions

**Comments to the Author**

1. If the authors have adequately addressed your comments raised in a previous round of review and you feel that this manuscript is now acceptable for publication, you may indicate that here to bypass the “Comments to the Author” section, enter your conflict of interest statement in the “Confidential to Editor” section, and submit your "Accept" recommendation.

Reviewer #2: (No Response)

2. Is the manuscript technically sound, and do the data support the conclusions?

Reviewer #2: Yes

3. Has the statistical analysis been performed appropriately and rigorously? 

Reviewer #2: Yes

4. Have the authors made all data underlying the findings in their manuscript fully available?

Reviewer #2: No

5. Is the manuscript presented in an intelligible fashion and written in standard English?

Reviewer #2: Yes

6. Review Comments to the Author

Reviewer #2: Dear authors

From my point of view, this manuscript is accepted to be published after all corrections

7. PLOS authors have the option to publish the peer review history of their article (what does this mean?). If published, this will include your full peer review and any attached files.

Reviewer #2: **Yes: **Dalia G Mahran

---

## [Editor Report · Acceptance letter]

17 Jul 2020

PONE-D-20-02630R2 

Functional outcomes after the treatment of hip fracture 

Dear Dr. Takahashi:

I'm pleased to inform you that your manuscript has been deemed suitable for publication in PLOS ONE. Congratulations! Your manuscript is now with our production department. 

Kind regards, 

on behalf of

Dr. Osama Farouk 

Academic Editor

PLOS ONE